# Combined Inhibitory Effect of Canada Goldenrod Invasion and Soil Microplastics on Rice Growth

**DOI:** 10.3390/ijerph191911947

**Published:** 2022-09-21

**Authors:** Xiaoxun Zhao, Hongliang Xie, Xin Zhao, Jiaqi Zhang, Zhiliang Li, Weiqing Yin, Aiguo Yuan, Huan Zhou, Sehrish Manan, Mudasir Nazar, Babar Iqbal, Guanlin Li, Daolin Du

**Affiliations:** 1School of Environment and Safety Engineering, Jiangsu University, Zhenjiang 212013, China; 2Ministry of Education Key Laboratory of Pollution Processes and Environmental Criteria, Nankai University, Tianjin 300350, China; 3Department of Civil and Environmental Engineering, College of Engineering, Seoul National University, Seoul 08826, Korea; 4Ministry of Education Key Laboratory for Ecology of Tropical Islands, Key Laboratory of Tropical Animal and Plant Ecology of Hainan Province, College of Life Sciences, Hainan Normal University, Haikou 571158, China; 5Zhenjiang Environmental Monitoring Center of Jiangsu Province, Zhenjiang 212004, China; 6Zhenjiang New District Environmental Monitoring Station Co., Ltd., Zhenjiang 212132, China

**Keywords:** *Solidago canadensis* L., alien plant invasion, microplastics toxicity, phenology, photosynthetic parameters, antioxidants enzyme

## Abstract

Alien plant invasion and residual soil microplastics (MPs) are growing threats to agricultural crop production. This study determined the adverse effects of Canadian goldenrod (*Solidago canadensis* L.) invasion and residual soil MPs on rice growth and development. The biomass, phenological indices, photosynthetic parameters, and antioxidant enzyme activities of rice were measured on the 50th and 80th day of post-plantation. Biomass and phenotypic results indicated the more harmful effects of the combination of *S. canadensis* invasion and residual soil MPs compared to *S. canadensis* invasion or residual soil MPs effects alone. Moreover, the interaction effect of *S. canadensis* invasion and residual soil MPs markedly reduced the ascorbate peroxidase and catalase belowground, while they increased in the aboveground parts of the rice. However, the *S. canadensis* invasion and residual soil MPs interactive treatments lowered the superoxide dismutase concentrations in the belowground parts of the rice plants while elevating the peroxidase and reactive oxygen species concentrations in both the belowground and aboveground parts compared to the other treatments. Among all treatments, *S. canadensis* invasion alone had the most negligible negative impact on rice biomass and growth indices. Our study suggests that soil MPs could negatively affect crop production with invasive alien plants, and the combined effects were more harmful than either of the single factors. Our findings will lay the groundwork for analyzing the impacts of invasive alien plants on rice crops.

## 1. Introduction

Alien plant invasion is considered a serious global environmental issue that is a growing threat to terrestrial ecosystem function and biodiversity by altering the composition and structure of aboveground and belowground biosensors [1,2,3,4]. The high ecological and economic expenses associated with alien plant invasions have inspired a wonderful pastime in elucidating how invaders impact terrestrial ecosystems, particularly agroecosystems. Several invasive alien plants have a wide range of adverse effects on crop production in invaded agroecosystem areas [5]. Invasive alien plants can inhibit germination and affect crop physiological activities, thereby reducing the quality and quantity of agriculture production through various mechanisms, including competing for resource (e.g., light, nutrients, and water), allelopathy, and parasitism, causing a continuously rising economic cost [6,7,8,9,10,11]. Thus, the possible consequences of alien plant invasion on agricultural production have been mentioned in experimental investigations, including a drastic reduction in yield [5]. Although extensive studies have been conducted to determine the relationship between alien plant invasion and crops that lead to such destructive outcomes, the detailed underlying process is still not clear [9,10,12].

The deposition of microplastics (MPs) in agricultural soil has received considerable attention owing to various factors such as eutrophication, wastewater supply, effluent addition, organic manure or agrochemicals, and mulch film use [13,14,15,16]. For example, sewage sludge and the degradation of plastic items account for over 80% of residual MPs and ingested by biota [17,18]. MPs are persistent anthropogenic pollutants that contaminate farmlands by altering the soil physicochemical properties [19,20,21,22,23], thereby affecting plant moisture absorption and nutrient delivery to crop root systems [24,25,26]. Meanwhile, the inhibition of seed germination, oxidative damage, and effects on belowground and aboveground biomass of plants caused by MPs have been identified as the most common side effects of residual MPs in the soil [13,15,27,28,29]. Furthermore, combining MPs and other environmental pollutants could considerably suppress crop growth by impacting photosynthesis and enzyme activities [20,22,30]. The presence of MPs in agroecosystem soils, particularly combined with other pollution and environmental issues, might pose significant environmental risks to the quality and quantity of crops as well as the health of human beings [31,32].

Photosynthesis, an essential process in plant life, is easily influenced by environmental stresses [22]. Moreover, photosynthesis, photorespiration, and respiration are all processes that produce reactive oxygen species (ROS) [33]. Under environmental stresses, the manufacturing of various ROS radicals primarily influences and conveniently triggers expressive oxidative stress of photosynthetic components, resulting in photoinhibition or photo-oxidation [34]. Antioxidant enzymes are mainly involved in reducing damage to cells and tissues due to excess ROS [32,33]. As emerging environmental stresses, both alien plant invasion and residual soil MPs in farmland can cause changes in the photosynthesis process and the production of ROS, which will further impact the production of crops. For example, the allelochemicals released from invasive alien plants and the uptake of MPs by plant roots might cause phytotoxicity. Consequently, allelochemicals secreted from the roots could inhibit crop respiration and photosynthetic activity, cause oxidative stress in target plants by inhibiting the antioxidant mechanism of the target species, and even cause phytotoxicity in the target species [10,11,12]. The delayed photosynthetic activity caused by MPs may contribute to the decrease in chlorophyll content in thale cress (*Arabidopsis thaliana* (L.) Heynh.) and maize (*Zea mays* L.) [35,36,37]. In contrast, the decreased photosynthetic efficiency was responsible for the unbalanced chlorophyll content in garden cress (*Lepidium sativum* L.) and lettuce (*Lactuca sativa* L.) [37,38]. Thus, the higher ROS production in belowground than aboveground parts might indicate severe damage to the crops [39]. Moreover, when exposed to MPs, the antioxidant enzymes (i.e., superoxide dismutase (SOD) and peroxidase (POD) activities were suppressed, and catalase (CAT) activity was reduced as well [22,40]. Since alien plant invasion and soil MPs are common and concomitant emerging environmental contaminants in agroecosystems, it is critical to evaluate the combined effects of alien plant invasion and soil MPs on crop production by measuring the photosynthetic parameters and antioxidant enzyme activity.

Canadian goldenrod (*Solidago canadensis* L.), a perennial herbaceous plant in the Asteraceae family, is native to North America. Due to their extreme growth pattern, characterized by rapid and plentiful germination, rapid development, and high reproductivity [41,42], this species has become one of the most damaging and widespread invasive plants around the world up to now [43]. Consequently, *S. canadensis* invasion has disrupted agricultural areas and caused a decrease in the quality and quantity of agricultural production [43]. The long history of agricultural cultivation and intensive pattern of agricultural activity have led to the vast and various residual MPs in farmland soils. Thus, agricultural crop production faced a new challenge induced by the *S. canadensis* invasion and residual soil MPs. Although many studies have investigated the impact of alien plant invasion or MPs on crop production, to the best of our knowledge, studies to determine the combined effects of alien plant invasion and MPs on the growth, photosynthetic, and physiological parameters of crops are still limited. To fill these knowledge gaps, we established an experiment in which rice (*Oryza sativa* L.) was grown in pots with polyethylene MPs or *S. canadensis* to determine the individual and interactive effects of *S. canadensis* invasion and MPs on rice growth, phenological indices, photosynthetic parameters, and antioxidant enzyme activities on two different days after transplanting. We hypothesized that (1) *S. canadensis* invasion and MPs could both negatively affect the crop morphological and physiological parameters; (2) *S. canadensis* invasion would have less effect on the biomass and physiological activities of the crop compared to MPs, although the interaction of *S. canadensis* invasion and MPs would have more significant negative impacts on the parameters above; and (3) the antioxidant enzyme activities in the belowground and aboveground parts would change due to the combined effect of *S. canadensis* and MPs because of *S. canadensis* growth interacting with the MPs present in the soil. Our findings provide novel evidence to understand the interactive effects of *S. canadensis* invasion and MPs on crops.

## 2. Materials and Methods

### 2.1. Materials Preparation

Given the broad application of agricultural mulch files, polyethylene MPs, a prominent plastic pollutant in the environment, were utilized in this study to stimulate MP residues in farm soils [31,44]. The MPs used in this study were polyethylene particle mixtures of pellets, fragments, and fibers, linked with various types of MP particles in the natural environment [45]. Pellet particles were purchased from Dongguan Zhangmutou Huahuang plastic material firm Dongguan, Guangdong China, and the experimental pellet MPs were obtained by sieving between 18-mesh (1.00 mm) and 30-mesh (0.60 mm) sieves. The experimental fragment MPs were obtained by hand cutting the polyethylene plate, purchased from Shanghai Dayou Hardware Co., Ltd., and then sieved using the same processes as the pellet particles. The experimental fiber MPs were obtained by manually cutting the polyethylene rope (diameter = 0.20 mm) into fiber particles (<2.0 mm). The three types of polyethylene MPs were mixed at a ratio of 3:4:3, the commonly detected mixture ratio of the three types of polyethylene MPs in the environment [46]. Before use, the polyethylene MPs mixture was successively rinsed with ethanol (70%) and deionized water to remove the solvent chemicals on its surface. Then, it was air-dried at room temperature [40]. All the MPs were stored at 4 °C before use.

Ning Japonica 12 hybrid rice seedlings (Ning 12) were obtained from the Baima Research Station of Nanjing Agricultural University in June 2021. The experimental *S. canadensis* seedlings were obtained by growing the seeds in a greenhouse in April 2021, which were collected from a successful *S. canadensis* invasion area in Zhenjiang City, Jiangsu Province, China, in December 2020. After 2 months of cultivation, *S. canadensis* seedlings of similar size were selected before use.

The experimental soil was collected from the surface soil (0–20 cm) of greenspace with no MPs or *S. canadensis* invasion at the Jiangsu University campus (32°12′ N, 119°30′ E), Zhenjiang City, in April 2021. After removing the plant residues and stones, the air-dried soil was sieved through a 10-mesh (2.00 mm) sieve and stored in the dark before use.

### 2.2. Pot Experiment Design and Plant Sampling

The pot experiment was conducted in a greenhouse located on the campus of Jiangsu University from June to October 2021. The experimental pot design was a complete factorial with four treatments and fourteen replicates (56 plots in total), consisting of a control treatment (CK) with neither *S. canadensis* nor MPs in the pots, *S. canadensis* invasion treatment (SI), residual soil MPs treatment (MPs), and combination of *S. canadensis* invasion and residual soil MPs treatment (SI × MPs). Thus, 3.8 kg of advance prepared soil was weighed and placed in an open container. Then, *S. canadensis* seedlings of identical size were transferred into pots with the *S. canadensis* invasion treatment, and sterilized MPs (polyethylene MPs), corresponding to 0.5% of the soil weight, were added to the soil with the residual soil MPs treatment. After two weeks of soil stabilization, two rice seedlings of identical size and two to three leaves were transferred into each pot containing a simulated dry field.

After planting, all pots were cultivated in the greenhouse under natural conditions. Plants were sampled twice during the experiment period, specifically at the 50th and 80th days after transplanting (DAT), respectively. Seven pots of each treatment were sampled at each sampling time.

### 2.3. Plant Growth and Biomass Parameters

At the 50th and 80th DAT, one plant of the rice of each treatment was used to record root, shoot, and total biomass, whereas the other rice plants were left for further identification. Before uprooting, all photosynthetic activities were observed in the same plant. The rice roots were washed before being immersed in a 20 mmol L^−1^ EDTA-2Na solution for 15 min to remove the MPs that adhered to the root surfaces. The roots were then rinsed with deionized water. The roots, stems, and other parts were dried at 105 °C for half a minute before being dried at 75 °C to a constant weight.

### 2.4. Chlorophyll Contents and Gaseous Exchange Parameters

The chlorophyll contents recorded as SPAD values were obtained from the three highest fully grown leaves on each plant at 50th and 80th DAT using a SPAD meter (SPAD-502, Minolta Camera Co., Osaka, Japan). Gaseous exchange parameters were calculated using a LI-3800 system (Li-COR Inc., Lincoln, NE, USA). To reduce the gradients between laboratory atmospheric conditions and the inside of the gas exchange system’s cuvette, the sample CO_2_ concentrations, moisture content, and leaf temperature were customized to 400 µmol mol^−1^, 65%, and 28 °C, respectively. Based on the light response curve data, the photosynthetic photon flux density within the LI-3800 cuvette was adjusted to 1500 µmol m^−2^ s^−1^ for the photosynthetic activity measurements. The conditions inside and outside of the chamber were similar, and the leaves were kept in the cuvettes till stable values within the 30 s. The ambient conditions inside the cuvettes were prepared as described above, and the results were recorded when steady gas exchange measurements were performed, which required approximately 20 min. The following parameters were measured: net photosynthetic rate (Pn), transpiration rate (Tr), stomatal conductance (Gs), photosynthetically active radiation (PAR), intracellular CO_2_ concentration (Ci), and water-use efficiency (WUE).

### 2.5. Antioxidants Enzyme Activities

Plants were carefully removed from the pots on the 50th and 80th days. The roots were cleaned with distilled water to remove any residual soil particles. The plants were cut into pieces according to their organs (i.e., roots and stems). Then, excess water was removed with filter sheets. Nearly 0.2 g of each sample of the belowground and aboveground parts were weighed, and then crushed in liquid nitrogen with 5 mL of a phosphate buffer solution, which was made by combining 0.05 M Na_2_HPO_4_ and 0.05 M NaH_2_PO_4_, with a pH of 7.8. Then the mixtures were centrifuged at 18,000 g for 15 min at 4 °C. After centrifugation, the supernatants were collected and placed in an ice bath to measure enzyme activity (ascorbate peroxidase (APX), CAT, POD, SOD, and ROS). Specifically, APX and POD activities were determined using ascorbic acid and guaiacol oxidation techniques, respectively. SOD and CAT activities were tested using commercial kits (Nanjing Jiancheng, China), according to the manufacturer’s instructions. ROS activity was determined using 3,3’-diaminobenzidine [47] and nitrotetrazolium blue chloride staining [48].

### 2.6. Statistical Analysis

Repeated measure analysis of variance (ANOVA) was used to test the main effects of *S. canadensis* invasion, soil MPs, and time, as well as their interactions on all parameters (biomass, photosynthetic parameters, and antioxidant enzyme activities of belowground and aboveground parts) at two sampling dates (the 50th and 80th DAT) during the study period. A one-way ANOVA followed by Fisher’s least significant difference test was used to test the effects of treatment on all parameters at each sampling date. Principal component analysis (PCA) was performed to identify differences in the rice crop’s biomass, photosynthetic parameters, and antioxidant enzymes among different treatments for each sampling dataset. Redundancy analysis (RDA) and a permutational multivariate ANOVA (PERMANOVA) were used to analyze differences in the rice biomass and photosynthetic parameters concerning antioxidants enzymes. All analyses were performed using R software version 4.1.1 (Vienna, Austria) [49]. 

## 3. Results

### 3.1. The Response of Rice Biomass and Phenological Indices to Treatments

The impacts of *S. canadensis* (SI) invasion and/or residual soil MPs caused substantial shifts in the belowground, aboveground, and total biomass production of rice crops compared with the control. Overall, *S. canadensis* invasion treatment suppressed rice growth; that is, *S. canadensis* invasion contributed to less plant biomass (including belowground, aboveground, and total biomass) compared to the control treatment. Residual soil MPs decreased plant biomass more than *S. canadensis* invasion, and the interaction of *S. canadensis* invasion and residual soil MPs showed the most damaging effects on rice plant growth. The results of the RMANOVA highlighted the significant impacts of *S. canadensis* invasion and/or residual soil MPs on the belowground, aboveground, and total biomass. Similarly, significant interaction effects were recorded for belowground and total biomass, whereas no significant effects were observed for aboveground biomass (Appendix A). More substantial reductions in belowground biomass (ca. 65.03% and 48.18%), aboveground biomass (73.48% and 67.28%), and total biomass (69.4% and 65.57%) at the 50th and 80th DAT were recorded, respectively (Table 1).

The phenotypic characteristic estimation results indicated that the interaction between the *S. canadensis* invasion and residual soil MPs decreased the stem height (SH), plant height (PH), diameter (DM), number of leaves (NOL), tillers (NOT), and nodes (NON) per plant compared to the control treatment. At the 50th DAT, all the above parameters were decreased. At the 80th DAT, the same NOL was observed in the control and *S. canadensis* invasion treatments, while a slight reduction was recorded for the total height and significantly for all other parameters. Moreover, the same NOL was detected in residual soil MPs and the interactive treatment. The presence of *S. canadensis* invasion and residual soil MPs interactive treatment decreased the SH, PH, DM, NOT, and NON, but not the NOL of the rice (Table 2). According to the RMANOVA, significant differences were found only in DM, whereas no significant differences were recorded for other parameters (Appendix A).

### 3.2. The Response of Photosynthetic Parameters to Treatments

To study the effects of *S. canadensis* invasion and residual soil MPs on the photosynthetic parameters of a rice, SPAD values and leaf nitrogen (LN) of the rice were measured. Generally, *S. canadensis* invasion and/or residual soil MPs treatment reduced the SPAD and LN. The interaction of *S. canadensis* invasion and residual soil MPs declined SPAD and LN mostly, followed by MPs treatments, and *S. canadensis* invasion treatments triggered the lowest reduction (Table 3). According to the RMANOVA, the SPAD values and LN revealed significant effects of *S. canadensis* invasion and residual soil MPs. In contrast, interactive effects were significant for the SPAD value, and no meaningful interactions for LN were observed during the growth period (Appendix A).

All treatments were attributed to significant changes in the net photosynthetic rate (Pn), transpiration rate (Tr), and stomatal conductance (Gs) compared to the control. The *S. canadensis* invasion treatments had the most negligible impact on the photosynthetic parameters compared to all other tested treatments. Contrary to the above results, Pn showed higher activity at the 50th DAT than the 80th DAT, whereas the opposite was true for Tr and Gs activities. Highly significant effects were observed for Pn, Tr, and Gs between DAT. Additionally, among the treatments, the interaction treatment (SI × MPs) caused more significant reductions in Pn, Tr, and Gs activities. All treatments reduced the plant photosynthetic parameters, and *S. canadensis* invasion showed the least influence. The interactive treatments contributed to the greatest reduction (ca. 59.08%, 76.82%, and 75.17% of Pn, Tr, and Gs activities at the 50th DAT, and 63.61%, 61.51%, and 77.15% at the 80th DAT, respectively) compared to the control treatment. Significant reductions were also observed in photosynthetically active radiation (PAR), intracellular CO_2_ concentration (Ci), and water-use efficiency (WUE) of *S. canadensis* invasion, soil MPs, and interactive treatments compared to the control treatment. The *S. canadensis* invasion or soil MPs treatments alone triggered a less significant reduction in PAR, Ci, and WUE as compared to the interactive treatment. The PERMANOVA showed highly significant effects caused by *S. canadensis* invasion or soil MPs treatment on PAR, Ci, and WUE, while a considerable interaction between *S. canadensis* invasion and soil MPs treatment for WUE was observed; no significant effects were observed for PAR and Ci (Table 3 and Appendix A).

### 3.3. The Responses of Rice Antioxidants Enzymes Activities to Treatments

Ascorbate peroxidase (APX) concentration in the belowground part of the rice showed the most significant decreases with the interactive treatment. In contrast, the aboveground part displayed a different pattern with the highest concentration in the control, while the lowest was recorded in the *S. canadensis* invasion at both DAT (Table 4). Both belowground and aboveground APX were reduced by *S. canadensis* invasion, residual soil MPs, and the interaction of *S. canadensis* invasion and residual soil MPs treatment compared to the control treatment. The lowest aboveground APX reduction was detected after the interaction treatment at the 80th DAT. The results revealed highly significant values for APX concentrations in the belowground and aboveground parts at both DAT (Table 4 and Appendix A).

The treatments in the belowground parts detected remarkable reductions in CAT content. A significant increase was observed at the 50th and 80th DAT in the aboveground parts compared to the control treatments. The CAT content in the belowground parts decreased considerably by 28.5%, 45.7%, and 56.8% at the 50th DAT, and 25.8%, 42.8%, and 58.4% at the 80th DAT in the *S. canadensis* invasion, residual soil MPs, and interaction between *S. canadensis* invasion and residual soil MPs treatment, respectively, compared to the control treatment. Moreover, the aboveground CAT content increased markedly by 51.3%, 120.5%, and 197.2% at the 50th DAT, and 37.4%, 112.8%, and 270.4% at the 80th DAT compared with that of the control treatment (Table 4 and Appendix A).

Compared to the control, the *S. canadensis* invasion and MPs treatment considerably decreased the POD content in the rice plants’ belowground parts at the 50th DAT, while the MPs treatment substantially increased the POD content at 80 DAT. Interaction treatments resulted in higher POD contents than the control treatment at both DAT. In the belowground parts, *S. canadensis* invasion and MPs treatments reduced the POD content (ca. 33.8% and 12.5%, respectively), while the interaction treatments increased the POD content (ca. 43.2%) compared to that of the control treatments (control) at the 50th DAT. Moreover, the *S. canadensis* invasion treatments decreased the POD content (ca. 28.3%), whereas MPs and interaction treatments elevated the POD content by 31.9% and 60.0% at the 80th DAT. However, contrary to these results, treatments contributed to opposite patterns in the POD content in the aboveground parts of the rice plants at both DAT (Table 4). The *S. canadensis* invasion, residual soil MPs, and interaction treatments increased (ca. 14.8%, 35.4%, and 73.6%, respectively) the POD content in the aboveground parts at the 50th DAT, but which was lowered (ca. 25.4%, 41.6%, and 55.1%) at the 80th DAT compared to the control, respectively (Table 4 and Appendix A).

All treatments decreased the SOD content in the belowground parts of the rice plants at both DAT compared to the control; however, the overall SOD contents at the 50th DAT was higher than those at the 80th DAT. *S. canadensis* invasion, soil MPs, and interaction treatments triggered a notable reduction in SOD content in both the belowground and aboveground parts of the rice plants at the 50th and 80th DAT compared with the control (Table 4).

Overall, *S. canadensis* invasion, soil MPs, and interaction treatments induced significant changes in the ROS content of the rice plants’ belowground and aboveground parts compared to the control treatment. Specifically, *S. canadensis* invasion, soil MPs, and interaction treatments lowered the ROS content in the belowground parts at the 50th DAT compared to the control treatment, while the opposite was true at the 80th DAT. However, *S. canadensis* invasion, soil MPs, and interaction treatments raised the ROS content in the aboveground rice crop parts (ca. 15.7%, 61.2%, and 129.0%) at the 50th and (ca. 32.0%, 61.3%, and 104.2%) 80th DAT compared to the control treatments (Table 4 and Appendix A).

The amplitude of the interactive changes between biomass, photosynthetic parameters, and antioxidant enzymes of the rice was represented by principal component analysis (PCA), here showing a two-dimensional PCA of all the properties, explaining 79% and 67% of the total variance for 50th and 80th DAT. The PERMANOVA showed significant differences (*p* = 0.001) between the plant parameters at the 50th and 80th DAT, respectively (Figure 1). Moreover, the redundancy analysis (RDA) between the antioxidant enzymes in the rice and biomass production at the 50th and 80th DAT is presented in Figure 2a,b, while the analysis of enzymes and photosynthetic parameters was recorded and is shown in Figure 2c,d. The correlation between crop photosynthetic parameters and enzyme activities in both parts is presented in heatmaps (Figure 3). The heatmaps show that changes in photosynthetic parameters also influenced the enzyme activity of both crop parts, and this change was DAT dependent.

## 4. Discussion

Understanding the effects of alien plant invasion and residual soil MPs on agricultural crop exploration and implementation is essential in agricultural frameworks. Thus, the adverse impact of invasive alien plant and/or soil MPs contamination in food webs pose high risks to human health and safety.

### 4.1. Effects of S. canadensis Invasion and Residual Soil MPs on Rice Biomass and Phenology

Our results suggested that both the individual and interactive effect of *S. canadensis* invasion and residual soil MPs significantly negatively affected the biomass of rice (Table 1 and Table 2, Appendix A). In particular, *S. canadensis* invasion caused the least reduction in rice biomass and phenological indices, while the combined effect of *S. canadensis* invasion and residual soil MPs caused the most considerable decrease compared to control treatments. These findings confirmed our first and second hypotheses.

*S. canadensis* invasion is documented as an ecological threat to crops and a few studies have demonstrated its impacts on crops; this may be due to the presence and release of toxic materials that induced minor changes in the soil physicochemical quality, but had higher interactive effects on crops [42,50,51]. These minor changes are interrelated to toxic materials present in the soil, which also induced a significant change in the application of various models related to plants in the soil [51]. Similarly, *S. canadensis* affects crops by changing the soil physicochemical properties due to higher secretion of allelochemicals from the roots and alternatively had negative effects on the crop’s biomass and phenological indices [52]. Moreover, MPs in the soil also have a significant influence on various crops’ biomass as they disturb the soil microenvironment and fertility and thus induce changes in the physiological activities of crops [13,20,28,29,31]. Although the interactions of *S. canadensis* invasion and residual soil MPs were previously unknown, this study demonstrated that they had a negative effect on biomass (Table 1 and Appendix A). It is worth noting that each treatment studied has a different effect on the crops, with the interactions of *S. canadensis* invasion and soil MPs having the most detrimental impact on crop biomass. These results might be due to the highly changing soil properties and biogeochemical cycle induced by *S. canadensis* that encountered MPs present in the soil, penetrating the root, and moving towards the aerial parts of the crops, indicating that they have a severely damaging impact on both crop parts [22,31,50,53,54]. It was also observed that the combination of *S. canadensis* invasion with MPs increased the toxicity level by causing the highest rice inhibitory activity at both DATs. In contrast, the decline in biomass was higher at the 80th DAT than the 50th DAT. These disparities may be due to the interactions of *S. canadensis* invasion and the soil MPs’ exposure to the rice and cultivation time. These results might be due to the low photosynthetic, antioxidant enzyme activity, and nutrient availability at the later growth stage, which resulted in low biomass compared to the early growth stage [55].

### 4.2. Effects of S. canadensis Invasion and Residual Soil MPs on Photosynthesis

The present study observed that the photosynthetic parameters of rice were significantly affected by the individual and interactions of *S. canadensis* invasion and soil MPs (Table 3 and Appendix A). Thus, the effect was more severe considering the interactive treatment than the individual treatments at both the 50th and 80th DAT. In this study, the SPAD values, leaf nitrogen, and photosynthetic parameters were markedly reduced in the leaves of rice growing in soil with *S. canadensis* invasion or residual soil MPs than in the control treatment. Previous studies have explained soil–plant systems and discovered that *S. canadensis* invasion or soil MPs reduced the SPAD values and photosynthetic parameters, which had a significant impact on the biomass of belowground and aboveground crop parts [9,13,52,56]. In addition to *S. canadensis* invasion or MPs alone, this study revealed that the interactive effects strongly impacted the crop photosynthetic parameters. Conversely, the SPAD values and leaf nitrogen showed no significant impact at the 50th and 80th DAT; however, the photosynthetic parameters had a highly significant effect at both DATs in the rice crop (Appendix A). These findings could be attributed to *S. canadensis* invasion, which has a substantial impact on the microbiological and physicochemical properties of the soil [52,53], thus inducing a more significant effect on the nutrient availability to the crops and also had more considerable competition for available resources [55], and could increase the MP toxicity by lowering the photosynthetic parameters, which could have a negative impact on crop quality, which is a concern that needs to be investigated further.

Moreover, *S. canadensis* increased the competition for sunlight and nutrients and had a negative impact on the crop photosynthetic parameters [43]. MPs in the soil have also been found to be responsible for lower photosynthesis and transpiration rates in crop leaves [13]. For the first time, we discovered that the interaction of *S. canadensis* invasion and residual soil MPs led to a significant reduction in SPAD values and photosynthetic parameters in plant tissues. The results indicate that stress inducers compete for soil nutrients and their translocation into the crop sink. The MP-driven rise in roots limited nutrient absorption and enhanced the impacts of *S. canadensis* invasion by inducing changes in the belowground parts, which had a strong impact on the aboveground parts of the crop, suggesting that *S. canadensis* invasion might have intensified the deleterious effects of MPs on the crops and elevated the stress response of combination treatment of *S. canadensis* invasion and residual soil MPs, which had strong negative effects on the SPAD values and photosynthetic parameters at the 50th and 80th DAT of the rice than in the control treatment.

### 4.3. Effects of S. canadensis Invasion and Residual Soil MPs on the Antioxidant Enzyme Activities

MPs strongly adsorb chemicals found in soil because of their hydrophobic surfaces, which influences their bioavailability and environmental behavior [35,57]. In this study, *S. canadensis* invasion and soil MPs changed the antioxidant enzyme activity; however, combination treatment had more detrimental effects on the belowground and aboveground parts of the rice crop (Table 4 and Appendix A). *S. canadensis* invasion and soil MPs caused substantial differences in the antioxidant enzyme activity (APX and SOD) in rice crops’ belowground and aboveground parts at both DATs. This might be because the *S. canadensis* invasion and soil MPs utilized in this investigation could be termed as severe environmental pollution and induce significant changes in the soil physicochemical properties and physiology of the crops [9,26,58,59]. Similarly, the crops displayed varied antioxidant enzyme activity in the belowground and aboveground parts of the rice because of high changes in environmental impacts [59]. This could be due to the counter-effects of *S. canadensis* invasion on the roots and their strong changes in soil properties and negative effects on the aboveground parts of the crops, which could influence the severity of the threat of MPs in the belowground system. The interaction of *S. canadensis* invasion and residual soil MPs increased APX and decreased SOD enzyme activity in the aboveground parts of the rice when compared to *S. canadensis* invasion or soil MPs alone (Table 4). Further, antioxidant enzyme activity, that is, APX, increases oxidative stress and decreases SOD activity to avoid cell damage, meaning the crop’s stress is balanced, resulting in a decrease in biomass. Reduced APX activity belowground demonstrated a direct effect of *S. canadensis* invasion and residual soil MPs due to oxidative stress, whereas increased APX activity aboveground confirmed MPs movement in the crop body, which has a negative effect on rice physiology. These results are in line with previous research that demonstrated how alien invasive species greatly alter the soil physicochemical properties and directly influence the crop’s belowground growth [53]. This belowground growth restricts the movement of macro- and micro nutrients to the crops, facilitates the MPs’ movement, and had a greater impact on the antioxidant activity in the crop body [22]. Furthermore, it might be due to changes in the soil biogeochemical cycle caused by *S. canadensis* invasion, whereas MP movement caused numerous changes in physiological functions, and their combination in this study resulted in a significant decrease in SOD activity when compared to the control. Similarly, SOD activity was higher at the 50th DAT than at the 80th DAT, which could be attributed to oxidative stress in the early stages in both organs.

The CAT enzyme showed no significant activity between different DATs in the belowground part, but highly significant activity was seen at the 50th DAT as compared to the 80th DAT in the aboveground part (Table 4). These findings could be explained by the fact that the reaction to oxidative stress is markedly higher in the aboveground crop parts than in the belowground parts. Recent studies have also shown that alien invasive plants induced changes in soil properties which directly affect the source–sink relationship of the crops and negatively affect the physiological functioning by influencing crop growth [4,42,53,60], whereas MPs may encourage crops to produce an excessive amount of antioxidant enzymes, resulting in oxidative damage and an imbalance in crop photosynthetic efficiency [59,61]. POD is a key enzyme found in crop cell walls and cytoplasm that catalyzes the hydrolysis of H_2_O_2_ to H_2_O to metabolize the crop under stress. Thus, unlike the control treatment, *S. canadensis* invasion reduced POD content and increased it during MPs and combination treatments (Table 4). However, the effects were slightly higher at the 80th DAT than the 50th DAT. Furthermore, the aboveground POD activity was higher in the combination treatment at the 50th DAT and lower at the 80th DAT. This might be due to the pollutant response, which causes severe oxidative damage during early crop growth. Previously, strong negative impacts of both *S. canadensis* invasion and residual soil MPs on POD activity were reported to be in line with our current findings [9,10,13,59,62], whereas the interactive effect of *S. canadensis* invasion and residual soil MPs still needs to be studied in the future.

Generally, ROS elimination during ambient conditions throughout the plant production remains balanced [63]. Thus, an increase in ROS stimulates the crop stress response and generates more toxic compounds that alter the crop’s physiological functions [13,59]. Modifications in ROS production were treatment-specific and varied widely depending on the combination of treatments (with a more significant effect at the interactive treatment), as well as the plant organ (Table 4). Additionally, antioxidant enzymes and ROS generation also induced changes in the photosynthetic parameters of the crops (Figure 2c,d). Thus, the effect of MPs on various plant components may help explain some of the variability in ROS generation in rice crops [36,37]. In this study, ROS production decreased at the 50th DAT and increased dramatically at the 80th DAT in the belowground parts but increased in the aboveground parts at both DATs (Table 4). This increase in aboveground parts could be due to *S. canadensis* invasion, which change the soil biogeochemical and physical properties and combines with MPs to penetrate the plant body through the root, which might obstruct the healthy progression of the rhizosphere and root hairs and reduce moisture retention and crop nutrition [4,27,52]. A direct toxic effect of different treatments on root growth and function was observed, implying that further ROS generation increase could be determined by limiting oxidative stress and increasing enzymatic activities in the crops’ belowground and aboveground parts [9,13,42,52,59,62], which supports our results. As a result, further research on crop resistance to *S. canadensis* invasion/MP exposure and a deeper exploration of the proposed pathways of *S. canadensis* invasion and residual soil MP stresses on crops, as well as the potentially influencing factors, are required.

## 5. Conclusions

The findings of this study showed that solitary and a combination of *S. canadensis* invasion and residual soil MPs treatments strongly influenced rice crop biomass, phenological indices, photosynthetic parameters, and antioxidant enzyme activities at the 50th and 80th DAT. Thus, the crop responded to the four treatment combinations tested, with the interaction between *S. canadensis* invasion and residual soil MPs being one of the most harmful and *S. canadensis* invasion alone being considered the least damaging. Furthermore, the invasion of *S. canadensis* alters the effect of MP disturbance in the belowground and aboveground parts of the crop. Moreover, single (*S. canadensis* invasion or MPs) and interactive treatments dramatically altered the biomass and photosynthetic parameters in the leaves due to the higher disturbance in antioxidant enzyme activities in the belowground and aboveground parts of the rice crop. Our findings prompted concerns regarding the probable harmful repercussions of *S. canadensis* invasion and residual soil MPs interaction pollution in croplands for production and commercialization, although the seeds of rice, that is, the panicle, have not yet been examined. More research is needed to explore the potential impacts of *S. canadensis* invasion on crops, and their interaction with other pollutants; the transmission of soil MPs into the food chain; the influence of the *S. canadensis* invasion and residual soil MPs’ interaction on crop productivity, using long-term field experiments; and the resulting implications for food safety and human health.

## Figures and Tables

**Figure 1 ijerph-19-11947-f001:**
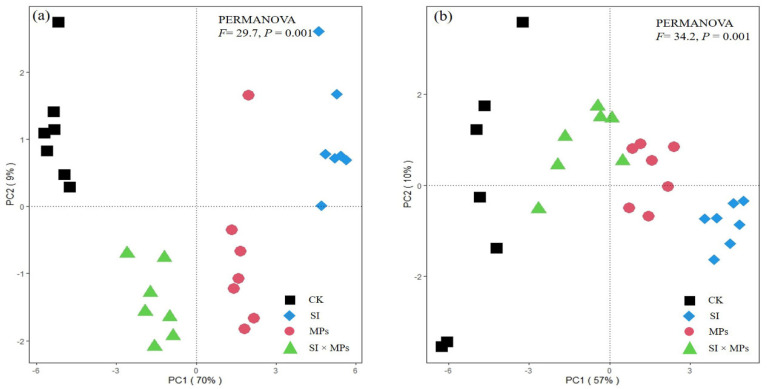
Principal component analysis (PCA) of all the biomass, photosynthetic parameters, and antioxidant enzymes of rice crop at the (**a**) 50th and (**b**) 80th days after transplanting. CK = control treatment; SI = *Solidago canadensis* L. invasion treatment; MPs = soil microplastics residual treatment; SI × MPs = combination of *S. canadensis* invasion treatment and soil microplastics residual treatment.

**Figure 2 ijerph-19-11947-f002:**
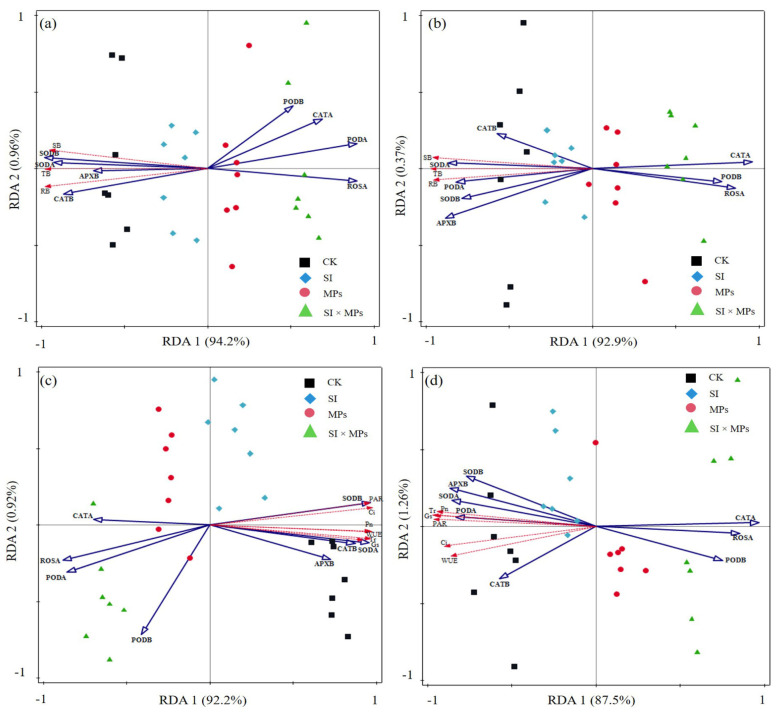
Redundancy analysis (RDA) of the antioxidant enzyme and biomass representing at the (**a**) 50th and (**b**) 80th days after transplanting, while the in-between antioxidant enzyme and photosynthetic parameters are represented at the (**c**) 50th and (**d**) 80th days after transplanting of the rice crop. CK = control treatment; SI = *Solidago canadensis* L. invasion treatment; MPs = soil microplastics residual treatment; SI × MPs = combination of *S. canadensis* invasion treatment and soil microplastics residual treatment; RB = root biomass; SB = stem biomass; TB = total biomass; Pn = net photosynthetic rate; Tr = transpiration rate; Gs = stomatal conductance; PAR = photosynthetically active radiation; Ci = intracellular CO_2_ concentration; WUE = water-use efficiency; Leaf N = leaf nitrogen; APXB = ascorbate peroxidase in belowground; APXA = ascorbate peroxidase in aboveground; CATB = catalase in belowground; CATA = catalase in aboveground; PODB = peroxidase in belowground; PODA = peroxidase in aboveground; SODB = superoxide dismutase in belowground; SODA = superoxide dismutase in aboveground; ROSB = reactive oxygen species in belowground; ROSA = reactive oxygen species in aboveground.

**Figure 3 ijerph-19-11947-f003:**
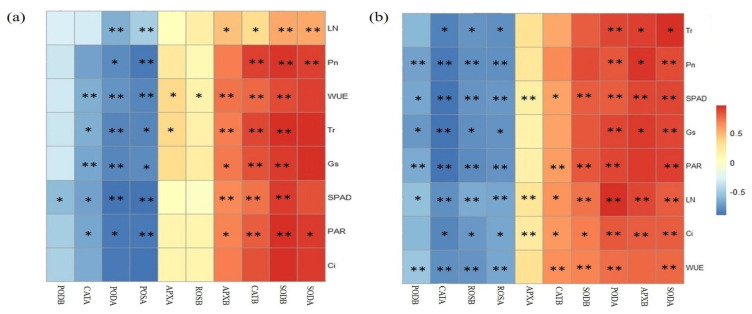
Heatmaps of the photosynthetic parameters and antioxidant enzymes of the rice crop: (**a**) 50th and (**b**) 80th days after transplanting. Asterisks indicate a highly significant (**) or significant (*) interaction between the different parameters. Pn = net photosynthetic rate; Tr = transpiration rate; Gs = stomatal conductance; PAR = photosynthetically active radiation; Ci = intracellular CO_2_ concentration; WUE = water-use efficiency; Leaf N = leaf nitrogen; APXB = ascorbate peroxidase in belowground; APXA = ascorbate peroxidase in aboveground; CATB = catalase in belowground; CATA = catalase in aboveground; PODB = peroxidase in belowground; PODA = peroxidase in aboveground; SODB = superoxide dismutase in belowground; SODA = superoxide dismutase in aboveground; ROSB = reactive oxygen species in belowground; ROSA = reactive oxygen species in aboveground.

**Table 1 ijerph-19-11947-t001:** Mean rice belowground, aboveground, and total biomass under different treatments at the 50th and 80th days after transplanting, presented as the mean ± standard error (*n* = 7).

Treatments		Belowground Biomass	Aboveground Biomass	Total Biomass
		(×10^−1^ g)	(×10^−1^ g)	(g)
50 DAT	CK	4.59 ± 0.10 a	5.26 ± 0.20 a	1.56 ± 0.02 a
	SI	3.72 ± 0.61 b	4.19 ± 0.06 b	1.23 ± 0.00 b
	MPs	2.62 ± 0.08 c	2.50 ± 0.12 c	0.78 ± 0.02 c
	SI × MPs	1.60 ± 0.08 d	1.40 ± 0.11 d	0.48 ± 0.01 c
80 DAT	CK	8.59 ± 0.07 a	24.94 ± 1.32 a	5.05 ± 0.08 a
	SI	6.78 ± 0.05 b	17.45 ± 0.95 b	3.52 ± 0.08 b
	MPs	5.27 ± 0.04 c	14.08 ± 0.51 c	2.78 ± 0.08 bc
	SI × MPs	4.45 ± 0.07 d	8.16 ± 0.23 d	1.74 ± 0.04 c

DAT = days after transplanting; CK = control treatment; SI = *Solidago canadensis* L. invasion treatment; MPs = soil microplastics residual treatment; SI × MPs = combination of *S. canadensis* invasion treatment and soil microplastics residual treatment. Different lowercase letters represent significant differences (*p* < 0.05).

**Table 2 ijerph-19-11947-t002:** Mean rice phenotypic indices at 50th and 80th days after transplanting, presented as mean ± standard error (*n* = 7).

Treatments		Stem Height	Total Height	Diameter	No. of Leaves	No. of Tillers	No. of Nodes
		(×10 cm)	(×10 cm)	(mm)			
50 DAT	CK	2.30 ± 0.10 a	6.10 ± 0.23 a	4.85 ± 0.34 a	10.86 ± 0.40	2.70 ± 0.30 a	8.40 ± 1.10
	SI	2.08 ± 0.08 ab	5.82 ± 0.12 ab	3.85 ± 0.24 b	9.29 ± 0.42	2.10 ± 0.30 ab	8.10 ± 0.40
	MPs	1.92 ± 0.05 bc	5.49 ± 0.19 b	3.65 ± 0.13 b	9.43 ± 0.48	1.70 ± 0.20 b	7.60 ± 0.30
	SI × MPs	1.70 ± 0.12 c	4.66 ± 0.17 c	3.36 ± 0.11 b	9.86 ± 0.34	1.60 ± 0.20 b	7.10 ± 0.30
80 DAT	CK	3.00 ± 0.06 a	6.49 ± 0.25 a	6.33 ± 0.26 a	14.00 ± 2.00	4.00 ± 0.30 a	10.70 ± 2.00 a
	SI	2.54 ± 0.10 b	5.99 ± 0.26 ab	4.97 ± 0.13 b	14.00 ± 1.00	3.00 ± 0.20 b	6.60 ± 0.40 b
	MPs	2.40 ± 0.09 b	5.57 ± 0.10 b	4.41 ± 0.12 c	12.00 ± 1.00	2.30 ± 0.30 bc	5.90 ± 0.60 b
	SI × MPs	1.99 ± 0.05 c	4.91 ± 0.17 c	3.84 ± 0.21 d	12.00 ± 1.00	1.90 ± 0.30 c	4.40 ± 0.50 b

DAT = days after transplanting; CK = control treatment; SI = *Solidago canadensis* L. invasion treatment; MPs = soil microplastics residual treatment; SI × MPs = combination of *S. canadensis* invasion treatment and soil microplastics residual treatment. Different lower-case letters represent significant differences (*p* < 0.05).

**Table 3 ijerph-19-11947-t003:** Mean rice photosynthetic parameters at the 50th and 80th days after transplanting, presented as the mean ± standard error (*n* = 7).

Treatments		Pn	Tr	Gs	PAR	Ci	WUE	SPAD	Leaf N
		(×10 μmol m^−2^ s^−1^)	(μmol m^−2^ s^−1^)	(×10^−1^ μmol m^−2^ s^−1^)	(×10^3^ μmol m^−2^ s^−1^)	(×10^2^ ppm)	(%)	(×10)	
50 DAT	CK	2.68 ± 0.11 a	6.23 ± 0.08 a	6.54 ± 0.16 a	1.55 ± 0.02 a	5.26 ± 0.06 a	9.96 ± 0.35 a	3.31 ± 0.08 a	2.87 ± 0.06 a
	SI	2.03 ± 0.05 b	3.62 ± 0.14 b	4.05 ± 0.16 b	1.32 ± 0.02 b	4.21 ± 0.02 b	5.91 ± 0.66 b	2.93 ± 0.03 b	2.70 ± 0.06 b
	MPs	1.43 ± 0.07 c	2.62 ± 0.07 c	2.41 ± 0.17 c	1.01 ± 0.02 c	3.18 ± 0.04 c	4.53 ± 0.06 c	2.63 ± 0.03 c	2.68 ± 0.07 c
	SI × MPs	1.10 ± 0.04 d	1.44 ± 0.13 d	1.62 ± 0.08 d	0.80 ± 0.02 d	2.13 ± 0.09 d	2.63 ± 0.17 d	1.89 ± 0.05 d	2.57 ± 0.08 d
80 DAT	CK	2.22 ± 0.04 a	7.22 ± 0.18 a	6.52 ± 0.12 a	1.40 ± 0.06 a	5.61 ± 0.16 a	7.80 ± 0.35 a	3.57 ± 0.06 a	3.35 ± 0.15 a
	SI	1.84 ± 0.01 b	5.48 ± 0.12 b	5.47 ± 0.10 b	1.08 ± 0.03 b	4.55 ± 0.14 b	5.77 ± 0.27 b	3.05 ± 0.06 b	2.80 ± 0.06 b
	MPs	1.21 ± 0.01 c	3.93 ± 0.19 c	3.52 ± 0.11 c	0.80 ± 0.02 c	3.33 ± 0.14 c	4.29 ± 0.30 c	2.48 ± 0.06 c	2.45 ± 0.03 c
	SI × MPs	0.81 ± 0.02 d	2.78 ± 0.05 d	1.49 ± 0.10 d	0.31 ± 0.05 d	2.64 ± 0.11 d	2.51 ± 0.23 d	1.63 ± 0.10 d	2.10 ± 0.05 d

DAT = days after transplanting; CK = control treatment; SI = *Solidago canadensis* L. invasion treatment; MPs = soil microplastics residual treatment; SI × MPs = combination of *S. canadensis* invasion treatment and soil microplastics residual treatment; Pn = net photosynthetic rate; Tr = transpiration rate; Gs = stomatal conductance; PAR = photosynthetically active radiation; Ci = intracellular CO_2_ concentration; WUE = water-use efficiency; Leaf N = leaf nitrogen. Different lowercase letters represent significant differences (*p* < 0.05).

**Table 4 ijerph-19-11947-t004:** Mean antioxidant enzyme activities and reactive oxygen species in the belowground and aboveground parts of the rice under different treatments at the 50th and 80th days after transplanting, presented as the mean ± standard error (*n* = 7).

Parts	DAT	Treatments	APX	CAT	POD	SOD	ROS
			(×10^2^ U min^−1^ g^−1^ FW)	(×10^2^ U min^−1^ g^−1^ FW)	(×10^3^ U min^−1^ g^−1^ FW)	(×10^2^ U g^−1^ FW)	(×10 U min^−1^ g^−1^ FW)
Belowground	50	CK	1.88± 0.19 a	5.00 ± 0.41 a	1.86 ± 0.14 b	4.67 ± 0.11 a	0.94 ± 0.04 a
		SI	1.24 ± 0.27 b	3.58 ± 0.12 b	1.23 ± 0.07 c	3.65 ± 0.10 b	0.71 ± 0.07 b
		MPs	1.11 ± 0.08 bc	2.72 ± 0.17 c	1.63 ± 0.21 bc	2.74 ± 0.11 c	0.65 ± 0.01 b
		SI × MPs	0.64 ± 0.10 c	2.16 ± 0.14 c	2.67 ± 0.10 a	1.61 ± 0.11 d	0.51 ± 0.02 c
	80	CK	2.86 ± 0.25 a	4.39 ± 0.95 a	3.26 ± 0.05 c	2.48 ± 0.20 a	1.40 ± 0.08 c
		SI	2.27 ± 0.15 b	3.26 ± 0.20 ab	2.33 ± 0.33 d	2.33 ± 0.04 a	1.78 ± 0.04 b
		MPs	1.49 ± 0.10 c	2.51 ± 0.35 b	4.30 ± 0.13 b	1.97 ± 0.05 b	2.08 ± 0.16 b
		SI × MPs	1.10 ± 0.07 c	1.83 ± 0.36 b	5.21 ± 0.18 a	1.48 ± 0.12 c	2.70 ± 0.13 a
Aboveground	50	CK	2.28 ± 0.21 a	9.05 ± 1.01 c	1.89 ± 0.08 c	8.14 ± 0.18 a	1.25 ± 0.04 b
		SI	1.03 ± 0.04 d	13.70 ± 1.44 bc	2.16 ± 0.13 c	5.43 ± 0.26 b	1.45 ± 0.08 b
		MPs	1.40 ± 0.13 c	19.97 ± 3.06 ab	2.55 ± 0.10 b	4.30 ± 0.21 c	2.02 ± 0.10 a
		SI × MPs	1.85 ± 0.10 b	26.90 ± 4.44 a	3.27 ± 0.74 a	3.59 ± 0.14 d	2.87 ± 0.16 a
	80	CK	3.95 ± 0.35 a	2.50 ± 0.10 d	4.36 ± 0.44 a	4.29 ± 0.17 a	1.40 ± 0.10 d
		SI	1.44 ± 0.46 c	3.44 ± 0.14 c	3.25 ± 0.23 b	3.31 ± 0.13 b	1.84 ± 0.60 c
		MPs	1.83 ± 0.08 c	5.32 ± 0.14 b	2.55 ± 0.11 bc	2.95 ± 0.27 b	2.25 ± 0.23 b
		SI × MPs	3.06 ± 0.17 b	9.26 ± 0.34 a	1.96 ± 0.08 c	2.36 ± 0.07 c	2.85 ± 0.08 a

DAT = days after transplanting; CK = control treatment; SI = *Solidago canadensis* L. invasion treatment; MPs = soil microplastics residual treatment; SI × MPs = combination of *S. canadensis* invasion treatment and soil microplastics residual treatment; APX = ascorbate peroxidase; CAT = catalase; POD = peroxidase; SOD = superoxide dismutase; ROS = reactive oxygen species. Different lowercase letters represent significant differences (*p* < 0.05).

## Data Availability

Not applicable.

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
