# Peer review of "Combined Inhibitory Effect of Canada Goldenrod Invasion and Soil Microplastics on Rice Growth"

_ijerph, 2022, doi:10.3390/ijerph191911947_

Round 1

Reviewer 1 Report

The authors addressed the effects of the interactive effects of Canada goldenrod invasion and soil microplastics on the growth, phenology and physiology of rice. The manuscript is illuminating an interesting aspect and has great potential. Both biological invasion and microplastics are hotspots in ecological and environmental research. The combined effects of them have rarely been investigated and therefore the study brings new information to the research area. The manuscript is well organized and well-written. I recommend the manuscript be accepted after addressing the following comments and errors.

1.       Introduction. The third hypothesis, ”the antioxidant enzyme activities in the belowground and aboveground parts would change”, is it caused by invasive plants or by the interaction between invasive plants and microplastics, or both? This is not clear.

Materials and Methods.

2.       On line 132, ”Pellet particles were purchased from Dongwan Zhangmutou 132 Huachuang plastic material firm”, is the city Dongguan or Dongwan?

3.       Do the chemicals on the surface of microplastics need to be washed off before the experiment?

4.       How is the relative size of rice and S. canadensis selected for the experiment determined?

5.       Why did the authors plant S. canadensis before rice in the experiment?

6.       Why there is only one S. canadensis in each pot? What is the basis?

7.       Latin names of species are not in italics.

Results

8.       In table 2, S. canadensis invasion treatment induced a significantly decrease in the diameter compared to the other treatments at the 50th DAT, but your statement is that “no significant variations were detected in the S. canadensis invasion treatment compared to the control treatment at the 50th DAT”.

9.       In table 3, the Leaf N in the S. canadensis invasion treatment was far greater than control treatment, which does not agree with your statement.

10.    The change of POD contents does not agree with your statement (on line 337 and 348).

11.    On line 365, what does “The amplitude of the interactive changes between biomass, photosynthetic parameters, and antioxidant enzymes of the rice at the 50th and 80th DAT, as represented by principal component analysis (PCA), where color combinations represent treatments concerning the control effectiveness” mean?

12.    The content of the figures is not shown in your results and discussion sections.

Discussion

13.    “S. canadensis invasion is documented as an ecological threat to crops; however, a few studies have demonstrated its impacts on crops because of minor changes in soil physicochemical qualities, but higher interactive effects on crops”. I can't understand what this sentence means.

14.    Why can S. canadensis invasion increase MPs toxicity by reducing photosynthetic parameters?

15.    Why do the authors measure these indicators at two DATs? In addition, why didn't the authors mention the significance of the difference between the two DATs in the first two sections of the discussion?

Reviewer 2 Report

I do find this work interesting and valuable, however there are some issues which should be clarified.

Line 38 – should be „rice”?

Line 254 – You wrote „…No significant variations were detected in the S. canadensis invasion
treatment compared to the control treatment at the 50th DAT (Table 2)”, But in Table 2  diameter  at 50DAT is marked as different comparing to control.

Table2 – there isn’t NON parameter there.

line 311– I suggest add „…showed the most significant decrease..” – beacuse all treatments showed significant decrease comparing to control

Lines 330 – 333 – mark please that this is about belowground parts

Lines 348-351 – which part of plant you refer to in this sentence?  Aboveground ? According to the Table 4 there is exactly opposite for aboveground parts: at 50 DAT POD is increasing in treatments and at 80DAT is decreasing compared to control.
